# Organizing recurrent network dynamics by task-computation to enable continual learning

**Lea Duncker**\*
Gatsby Unit, UCL
London, UK
duncker@gatsby.ucl.ac.uk

**Laura N. Driscoll**\*
Stanford University
Stanford, CA
lndrisco@stanford.edu

**Krishna V. Shenoy**
Stanford University
Stanford, CA
shenoy@stanford.edu

**Maneesh Sahani**[†]
Gatsby Unit, UCL
London, UK
maneesh@gatsby.ucl.ac.uk

**David Sussillo**[†]
Google Brain, Google Inc.
Mountain View, CA
sussillo@google.com

## Abstract

Biological systems face dynamic environments that require continual learning. It is not well understood how these systems balance the tension between flexibility for learning and robustness for memory of previous behaviors. Continual learning without catastrophic interference also remains a challenging problem in machine learning. Here, we develop a novel learning rule designed to minimize interference between sequentially learned tasks in recurrent networks. Our learning rule preserves network dynamics within activity-defined subspaces used for previously learned tasks. It encourages dynamics associated with new tasks that might otherwise interfere to instead explore orthogonal subspaces, and it allows for reuse of previously established dynamical motifs where possible. Employing a set of tasks used in neuroscience, we demonstrate that our approach successfully eliminates catastrophic interference and offers a substantial improvement over previous continual learning algorithms. Using dynamical systems analysis, we show that networks trained using our approach can reuse similar dynamical structures across similar tasks. This possibility for shared computation allows for faster learning during sequential training. Finally, we identify organizational differences that emerge when training tasks sequentially versus simultaneously.

## 1 Introduction

Computations in the brain are thought to be implemented via the dynamical evolution of activity within large populations of neurons. This dynamical systems view has offered insight into computational mechanisms underlying motor control [1], decision making [2–4] and timing tasks [5]. In these studies, animals are trained to perform a single task and neural population activity is typically recorded only after learning. Little is known about how a single neural population can learn to perform multiple computations without interference or forgetting, as new tasks are learned in sequence.

Based on recent experimental results on neural population subspace structure [6–8], we hypothesize that a careful organization of population dynamics into either orthogonal or shared subspaces may provide a robust implementation of multi-task computations that could also prove beneficial for sequential task learning. Similar tasks could evolve under similar dynamics in a shared subspace, while

---

[†]Equal Contribution

dissimilar tasks, or dissimilar components of tasks, could be confined to orthogonal subspaces. This organization would limit interference across unrelated computations and could allow for dynamics within each subspace to be learned independently.

To test and refine these ideas, we turn to training and analysis of recurrent neural networks (RNNs). Guided by the biological solution, we develop a novel algorithm for continual multi-task learning in RNNs. Our approach is based on a modification of the stochastic gradient descent (SGD) update to the network weights. Our learning rule aims to preserve network dynamics within subspaces used for previously learned tasks, and encourages interfering dynamics to explore orthogonal subspaces when learning new tasks.

We demonstrate our approach on networks trained to perform multiple tasks akin to those studied by neuroscientists in animal models [9]. Our proposed learning algorithm outperforms previous weight-regularization-based continual learning approaches on these tasks. We show that our learning rule encourages networks to utilize nearly orthogonal subspaces across tasks with opposite stimulus-response relationships that would otherwise interfere, but share subspaces for computations on similar tasks. Shared structure across similar tasks facilitates faster learning during sequential training. We highlight key differences in task alignment for networks trained under our sequential approach compared with those trained simultaneously on all tasks.

Ultimately, developing better approaches for continual learning in RNNs will provide new tools to investigate the organization of dynamics across tasks, and will aid the development of hypotheses about and comparisons with multi-task computation in the brain.

## 2    Background

### 2.1    Multi-task learning and catastrophic interference

Multi-task learning requires a single system to learn from example input/output pairs that reflect different task-specific relationships. In the simultaneous training setting, data from all tasks are interleaved within each minibatch used for training. In the sequential training setting, in contrast, tasks are introduced one at a time and are not repeated. Thus, once learning moves on to a new task, training data from earlier tasks become inaccessible. The key difficulty with sequential training is the problem of *catastrophic interference* or *forgetting* across tasks [10–12]. When data related to different tasks arrive sequentially, the weight updates required to improve performance on a new task can lead to changes in the network that impair performance on previously learned tasks. Since older tasks are never revisited, the network fails to achieve performance comparable to that attained during simultaneous training. Furthermore, the order in which tasks are presented may affect how well networks learn. Overcoming these challenges has been the subject of *continual learning* [11–15].

### 2.2    Related work

Previous work on continual learning in neural networks has predominantly focused on feedforward architecture. Previous approaches in this setting include regularisation of weight changes [13, 14], modification to the learning signal [15–17], network architecture modification [18, 19] or memory replay [20–22]. The algorithm for feedforward networks in [16] is most similar to our novel RNN proposal, and allows network weight changes only in directions orthogonal to the space spanned by previously seen inputs.

The problem of continual learning is less well-studied in recurrently connected networks [11], where unique challenges arise associated with continual learning in dynamical systems. Recurrent activity may feed back into the network over time possibly interfering in interactions with external inputs, or activity patterns of previously encountered tasks. Small individual weight changes can therefore have large effects on the evolution of the hidden state over time in the trial. Consequently, the approach in [16] does not directly extend to RNNs and [13] achieved only limited success in the RNN setting [9]. Recently, approaches to continual learning based on dynamical network architectures have also been developed in the RNN setting [23, 24]. For later comparisons, we focus on the approaches in [13, 14], which penalize changes in network weights deemed important for previous tasks, without modifying the network architecture during training. In both approaches, an importance measure for each network parameter is used as the weight of a quadratic regularisation term that limits changes

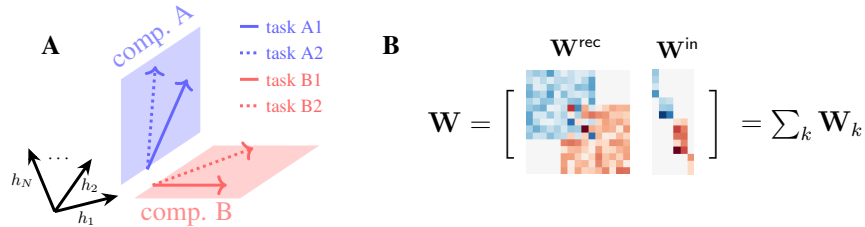

Figure 1: Multi-task computation via independent low-rank dynamics. **A:** Schematic of using orthogonal subspaces to implement different network computations. Tasks of the same class share computations (A or B) and therefore evolve in shared subspaces, while tasks across interfering task classes evolve in orthogonal subspaces. **B:** $\mathbf{W}$ is composed of different low-rank components $\mathbf{W}_k$. Non-interference depends on the orientation of singular vectors of each low-rank component.

to that parameter. These regularization terms are added to the overall objective function to limit interference.

## 3 Multi-task computation in recurrent networks

We consider an RNN, where hidden-state activity $\mathbf{h}$ evolves in time according to

$$\mathbf{h}_{t+1} = \phi(\mathbf{W}^{\text{rec}}\mathbf{h}_t + \mathbf{W}^{\text{in}}\mathbf{x}_t + \boldsymbol{\xi}_t) \tag{1}$$

with external input $\mathbf{x}_t$, uncorrelated Gaussian noise $\boldsymbol{\xi}_t$, and activation function $\phi$. The network activity is read out via the linear mapping

$$\mathbf{y}_t = \mathbf{W}^{\text{out}}\mathbf{h}_t. \tag{2}$$

Given the external input, and starting from an initial state, the network is trained to produce a target output timeseries $\mathbf{y}_t^*$.

Evidence from neural population data suggests that dynamical rules confined to orthogonal subspaces could serve as structures akin to independent modules, allowing a single network to perform different computations depending on contextual information or variable task demands [2, 25] (Figure 1A). In RNNs, an analogous solution could be implemented through structure in the network connectivity $\mathbf{W} = [\mathbf{W}^{\text{rec}}\ \mathbf{W}^{\text{in}}]$. $\mathbf{W}$ could be composed as a sum of low-rank dynamics matrices (Figure 1B), each structured to only produce output, or be sensitive to inputs along specific directions in high-dimensional space. Such low-rank dynamical components have been shown to successfully implement complex cognitive tasks [26, 27]. Within this framework, orthogonal structure across the input/output space of the different components could be used to limit interference along dimensions implementing unrelated computations. This would allow for network computations to be performed independently and be reused across tasks. Orthogonality across different dynamical components may also aid continual learning, since dynamics in orthogonal subspaces could be learned independently.

This organization of the network dynamics is desirable, yet arriving at it through sequential training is challenging. There is no reason to expect this structure to emerge without a constraint on the learning rule or network connectivity. At the same time, explicit parameterization would require constraints on the eigenstructure of the connectivity matrices, as well as setting the number and rank of the different dynamical components *a priori* – all of which is difficult in practice.

## 4 Continual learning through dynamical noninterference

We propose a novel continual learning algorithm that encourages networks to organize dissimilar dynamics into orthogonal subspaces. Consider activity patterns $\mathbf{z}_t^{k,r} = [\mathbf{h}_t^{k,r\ \mathsf{T}}\ \mathbf{x}_t^{k,r\ \mathsf{T}}]^{\mathsf{T}}$ at time $t$, on trial $r$ of task $k$. The $\mathbf{z}_t^{k,r}$ span the space of (external and recurrent) inputs processed by the network, while $\mathbf{W}\mathbf{z}_t^{k,r}$ span the space of recurrent output activations (prior to nonlinearity $\phi$) that the network has generated. In the standard SGD update, each change $\Delta\mathbf{W}$ in the weights is proportional to the gradient of the loss function, $\nabla_{\mathbf{W}}\mathcal{L}$. Interference from updates based on a new task arises when the singular vectors of $\Delta\mathbf{W}$ project significantly into the relevant input/output space of the previously

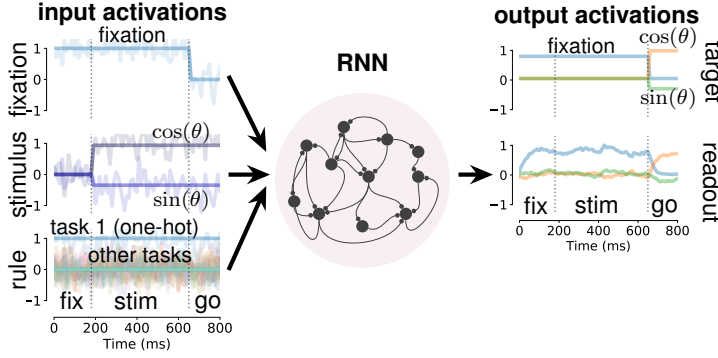

Figure 2: Schematic example of the `DelayPro` task. Left: noisy fixation, stimulus & rule input time-series. Input signal overlayed without noise for clarity. Right: target output activation and example readout of a trained network. Vertical dotted lines indicate task periods: fixation, stimulus & go.

learnt $\mathbf{W}$. Conversely, interference may be attenuated by reducing this overlap in input and output spaces. Thus, we propose a modification of the weight update that removes directions in the row and column spaces of $\Delta\mathbf{W}$ that align with the spaces spanned by $\mathbf{Z}_{1:k}$ (input) and $\mathbf{WZ}_{1:k}$ (output), respectively, where $\mathbf{Z}_{1:k} = [\mathbf{z}_1^{1,1}, \dots \mathbf{z}_T^{k,R}]$. Specifically, we project orthogonally to the relevant spaces using right- and left- multiplication by matrices

$$\mathbf{P}_z^{1:k} = \mathbf{I} - \mathbf{Z}_{1:k}(\mathbf{Z}_{1:k}^\top\mathbf{Z}_{1:k} + \alpha\mathbf{I})^{-1}\mathbf{Z}_{1:k}^\top = (\alpha^{-1}\mathbf{Z}_{1:k}\mathbf{Z}_{1:k}^\top + \mathbf{I})^{-1} \tag{3}$$

$$\mathbf{P}_{wz}^{1:k} = \mathbf{I} - \mathbf{WZ}_{1:k}(\mathbf{Z}_{1:k}^\top\mathbf{W}^\top\mathbf{WZ}_{1:k} + \alpha\mathbf{I})^{-1}\mathbf{Z}_{1:k}^\top\mathbf{W}^\top = (\alpha^{-1}\mathbf{WZ}_{1:k}\mathbf{Z}_{1:k}^\top\mathbf{W}^\top + \mathbf{I})^{-1}. \tag{4}$$

The small constant $\alpha$ ensures regularisation and invertibility; and the matrix inversion lemma yields the final expressions in terms of the covariance $\mathbf{\Sigma}_{1:k} = \mathbf{Z}_{1:k}\mathbf{Z}_{1:k}^\top$. This activity covariance matrix may be stored and incremented online after each task is learnt, allowing the corresponding projection matrices to be computed without access to previous activity patterns. When learning the $(k+1)$th task, we use the modified learning updates

$$\Delta\mathbf{W}_{\mathsf{CL}} \propto \mathbf{P}_{wz}^{1:k} \ (\nabla_{\mathbf{W}}\mathcal{L}) \ \mathbf{P}_z^{1:k}, \quad \Delta\mathbf{W}^{\mathsf{out}}_{\mathsf{CL}} \propto \mathbf{P}_y^{1:k} \ (\nabla_{\mathbf{W}^{\mathsf{out}}}\mathcal{L}) \ \mathbf{P}_h^{1:k} \tag{5}$$

where we have followed analogous definitions for the readout weights update.

Applying the projection matrices on both sides of the learning update prevents interfering weight changes along directions that span the activity space of previous tasks. Specifically, $\mathbf{P}_z^{1:k}$ projects away interfering directions in the input space of the weight update, while $\mathbf{P}_{wz}^{1:k}$ removes interfering directions of the output space. This means that large changes can now only occur in directions orthogonal to subspaces already used for computation, preserving the dynamics of previously learned tasks. In this way, our learning algorithm ensures that activity patterns generated in response to inputs on previous tasks are largely preserved throughout learning of new tasks. Computations that require dissimilar dynamics to those that were previously learned explore new, orthogonal subspaces. However, it is important to note that this learning approach does not automatically force all tasks to explore only orthogonal subspaces. If the new task dynamics don't interfere with a previously learned task (i.e. both task computations require similar dynamics), both tasks may evolve in shared activity-subspaces. Further details may be found in the supplementary material.

## 5  Results

We demonstrate our continual learning approach on a set of tasks previously used for studying multi-task representations in RNNs [9]. We chose four tasks to highlight how our proposed learning rule works on tasks with similar or opposed stimulus-response relationships. A delayed response task, `DelayPro`, can be divided into three periods, *fixation*, *stimulus* and *go*. The network receives a noisy fixation input and rule input at the beginning of the *fixation* period and throughout the trial. During the *stimulus* period, the network receives an angle input through two channels ($\sin\theta$ and $\cos\theta$) with added noise. The *go* period begins when the fixation input goes to zero, signaling the network to reproduce the input signal on the output channels. Figure 2 illustrates this task structure for an example `DelayPro` trial. In a related task, `MemoryPro`, the input stimulus disappears during an additional *memory* period before the go cue. In both `Pro` tasks, the network must respond in the same direction as the stimulus. In two tasks with the opposite stimulus-response relationship, `DelayAnti` and `MemoryAnti`, the network must respond in the opposite direction as the stimulus input.

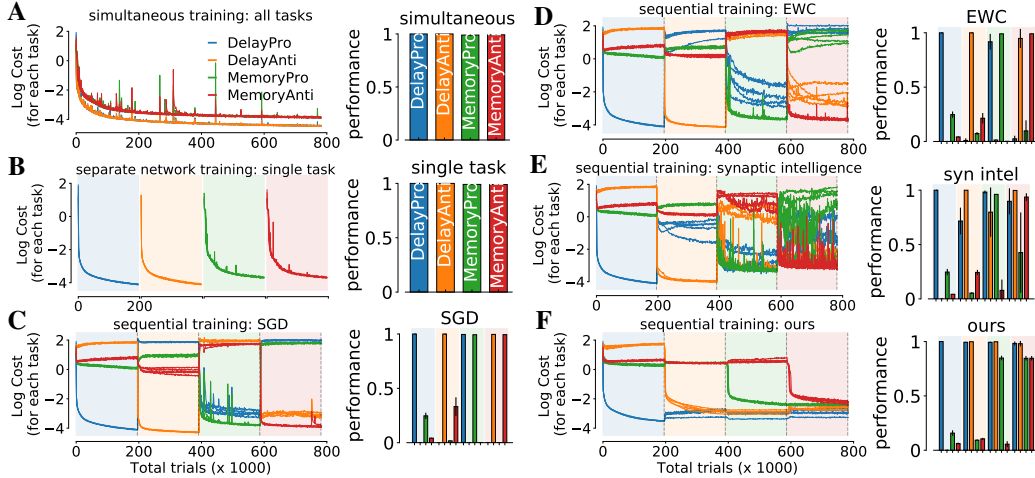

Figure 3: **A:** Left: log cost on test trials for each task throughout simultaneous training on all tasks. Right: performance on each task at the end of training. **B:** same as **A** but for separate networks trained on single tasks. **C:** Left: log cost on test trials throughout sequential training. Right: mean performance on each task at the end of each task's training sequence. Color shading indicates which task was trained for each epoch. **D, E, F:** Same as **C** using the continual learning approaches of [14] (**D**), [13] (**E**), and our approach (**F**). Loss curves in **A**-**F** are shown for 5 networks (different seeds).

Networks with rectified-linear activation functions were trained on these tasks to minimize the squared error between readouts and target outputs under added $L_2$-norm regularization of network weights and activity. Further details and results are provided in the supplementary material.

## 5.1 Comparisons with other continual learning approaches

We investigated whether our continual learning algorithm allows RNNs to learn multiple tasks in sequence, and how this compares to other methods. Figure 3 shows the log cost on test trials and task performance (following [9]) of networks trained with different learning approaches. In Figure 3A,B networks were trained either simultaneously on all tasks or separately on single tasks, providing an upper bound on the performance of any continual learning approach. Figure 3C shows the performance of networks trained sequentially on multiple tasks using unmodified SGD. The performance on previous tasks quickly degraded after new tasks with opposite stimulus-response relationships were introduced, illustrating the problem of catastrophic interference. Both Elastic Weight Consolidation (EWC) [14] and the synaptic intelligence algorithm [13] failed to maintain high performance on all tasks during learning (Figure 3D,E). By contrast, networks trained using our proposed continual learning algorithm maintained high performance on all previously learned tasks throughout training on later tasks (Figure 3F).

## 5.2 Training order effects on learning

We next investigated effects of training order on learning. The rate of decrease in the test error over learning was faster when networks were previously trained on a task with the same stimulus-response relationship (i.e. both `Pro` tasks or both `Anti` tasks) (Figure 4). This suggests that transfer learning across tasks with the same stimulus-response relationships may occur during sequential training. To better understand these training order effects, in the following sections we analyze the dynamics in trained networks.

## 5.3 Revealing task computations through fixed point structures

To better understand how learning similar tasks contributes to faster learning (Figure 4), we study how each task computation is performed. This is done by analyzing linearized dynamics around fixed points of the network for each task-period [28–30]. We examine dynamics in task specific subspaces identified through Targeted Dimensionality Reduction (TDR), a method to identify dimensions that

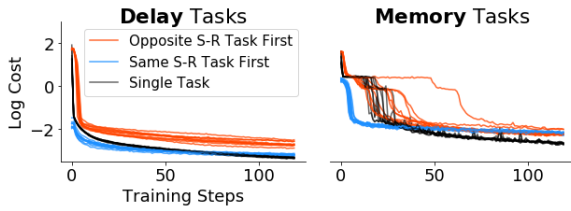

Figure 4: Log-cost during training of the second task, when the first task either had the same stimulus-response (S-R) relationship (blue) or an opposite one (orange) compared to training on one task only (black).

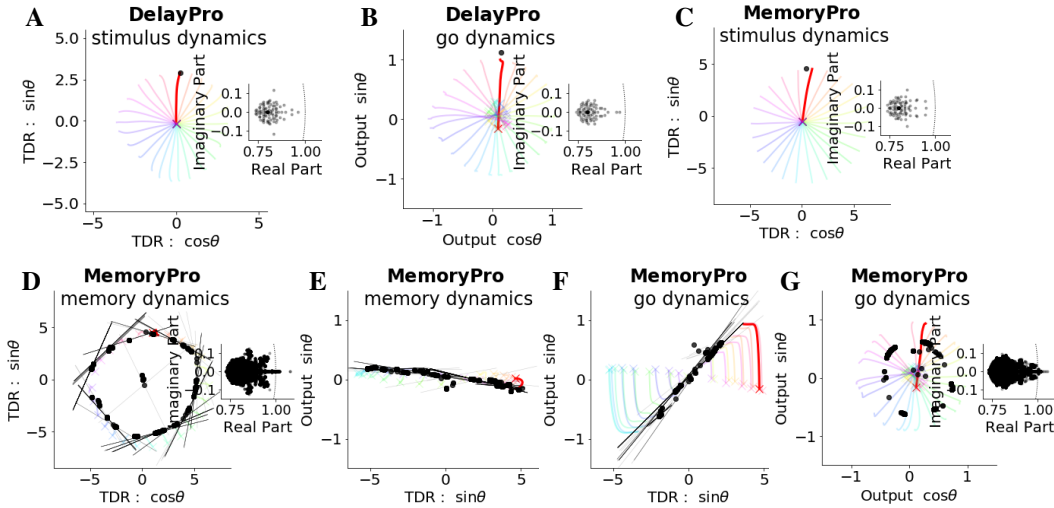

Figure 5: Fixed points(black) and hidden state activity(red) for a single trial, $\theta = \frac{\pi}{2}$. Activity from other trials of the same task are faded (colored by $\theta$) emanating from 'x'. Axes identified through targeted dimensionality reduction (TDR) or using vectors of $\mathbf{W}^{out}$ (Output). Insets are overlaid eigenspectra of all period specific fixed points. **A-B:** `DelayPro` stimulus, go period. **C:** `MemoryPro` stimulus period. **D-E:** `MemoryPro` memory period ring attractor in two subspaces. **F-G:** `MemoryPro` go period ring attractor in two subspaces. **D-G:** Unstable right eigenvectors of linearized dynamics around each fixed point are projected into activity subspace (black lines).

capture maximal variance in relation to a variable of interest [2]. We identify x and y axes that capture maximal variance of the hidden state across the two stimulus input variables $\cos\theta$ and $\sin\theta$ respectively (Figure 5). We also visualize go period dynamics in output space, vectors of $\mathbf{W}^{out}$.

On `Delay` tasks, a stimulus dependent stable fixed point emerges at stimulus onset. The hidden state evolves toward this fixed point in a subspace that is orthogonal to the readout (output null [6]). The go period starts when the fixation input goes to zero. Continued presence of the stimulus sets up another stable stimulus dependent fixed point in an output potent [6] subspace (Figure 5B).

`Memory` task stimulus period dynamics are equivalent to `Delay` tasks (Figure 5C). However, the stimulus eventually disappears and the network must retain information about the stimulus in the absence of input. To achieve this, a ring attractor emerges during the memory period (Figure 5D). The hidden state approaches the ring along stable dimensions and does not drift along unstable dimensions tangent to the ring. Without the stimulus input, go period dynamics rely solely on information in the hidden state to produce the appropriate readout. The network achieves this by setting up another ring attractor, oriented in an output potent subspace. The hidden state evolves toward this go period ring, maintaining the stimulus representation (Figure 5F-G). Fixed point structures across `Pro/Anti` tasks were similar, but with rotated relationships to output dimensions.

Once fixed point structures were learned for a particular task, there was minimal change to those structures during subsequent learning. All examined subspace angles between fixed point structures before and after learning subsequent tasks were less than 0.1 radians. Such minimal change is consistent with our learning rule limiting change of dynamics within previously explored subspaces.

## 5.4 Orthogonal and shared dynamics across tasks

We next identified the relationship between fixed-point structures across tasks to determine whether shared task representations could contribute to faster learning. We found stimulus-period fixed-point structures were mostly aligned across pairs of tasks with the same stimulus-response relationship and nearly orthogonal across `Pro` vs. `Anti` tasks (Figure 6A,B). Figure 6C shows the two ring attractors for go period dynamics in `MemoryPro` and `MemoryAnti` tasks are not aligned. Both rings are oriented such that they have a large projection onto readout axes, but are not aligned with each other. Initial conditions for `MemoryPro` go period dynamics (designated as 'x') are oriented nearly orthogonal to the ring of go period fixed points for the `MemoryAnti` task (Figure 6C). Our learning rule organized these orthogonal subspaces for go period dynamics across tasks with the opposite stimulus-response relationship to prevent interference across tasks. At the same time, tasks with the same stimulus-response relationship were evolved in the same subspace because their dynamics did not interfere with each other. Subspaces capturing stimulus-related variance are more aligned for tasks with the same stimulus-response relationships than those with the opposite relationship (Figure 6D). These results suggest that `Delay` and `Memory` tasks evolved in an aligned subspaces when they shared the same stimulus-response relationship. In these shared subspaces, reuse of dynamical structures that we identified in Figure 5 could explain transfer learning results from Figure 4.

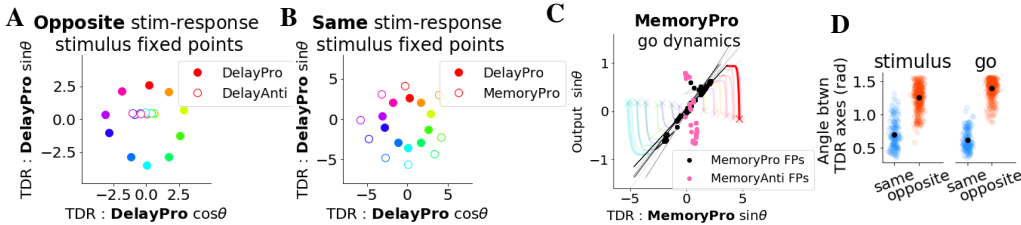

Figure 6: Subspace alignment after sequential training. Stimulus period fixed points for 10 trials spanning [0,2π) colored by θ **A**: `Delay`- **B**: `-Pro` tasks. **C**: `MemoryAnti` go period fixed points overlaid with `MemoryPro` go period fixed points and hidden state activity. **D**: Subspace angles (rad) between TDR axes for tasks with same (blue) or opposite (orange) stimulus-response relationships during stimulus (left) and go period (right). Each dot represents the angle between $\sin\theta$ or $\cos\theta$ TDR axis for a pair of tasks in one network (repeated for 9 task orders and 5 seeds).

## 5.5 Testing shared dynamics using input and hidden-state perturbations

We use perturbation experiments to further explore the possibility that similar tasks reuse dynamical structures in shared subspaces. First, we asked whether shared structure between `DelayPro` and `MemoryPro` tasks would enable them to perform similar computations even when inputs are inconsistent with those shown during training. Figure 7A shows the performance on the `MemoryPro` task when the network expected a memory period (`MemoryPro` rule input) or not (`DelayPro` rule input). Networks are able to perform memory-period dynamics even when the rule input did not instruct them to do so. Networks trained to perform the `MemoryPro` task first performed better, suggesting secondary learning of `DelayPro` might be reusing structure already present in the network. This phenomenon could explain faster learning apparent in Figure 4. Second, we ask whether the hidden state of `Pro` tasks contains information about `Anti` tasks and vice versa. Figure 7B shows task performance for networks where dimensions capturing 95% variance of a different task are projected out from the hidden state as the activity evolves in time. This more severely impaired the performance for tasks with the same stimulus-response relationship than with the opposite. These results show that computations rely on network activity patterns in subspaces that are shared between similar tasks.

## 5.6 Simultaneously trained tasks share output structure

We have shown that arranging network dynamics into computation-related subspaces minimizes interference during sequential training and allows transfer learning across similar tasks through reuse of dynamical structures. Yet, we found networks that were trained simultaneously on all tasks

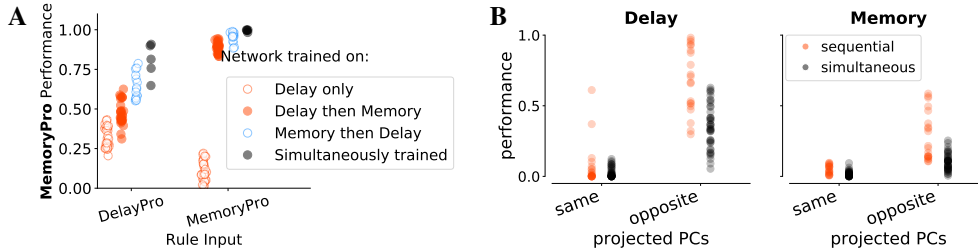

Figure 7: Input and hidden-state perturbations. **A**: Performance on `MemoryPro` given `DelayPro` or `MemoryPro` rule input under different training orders. **B**: Performance on one task when the dimensions capturing 95% of variance on another task have been projected out of the hidden state evolution either with the same or opposite stimulus-response relationship, plotted for 5 networks under an example training task order, and 10 simultaneously trained networks (see section 5.6).

achieved slightly better performance (Figure 3A). We therefore investigated organizational differences between sequentially and simultaneously trained networks.

Perturbation results from section 5.5 provide initial evidence that more features are shared across tasks in simultaneously trained networks. `DelayPro` trials could more successfully maintain stimulus information during a memory period (Figure 7A) and performance was more impaired when hidden-state activity in one task subspace was projected out from activity during another task (Figure 7B).

Indeed, fixed-point structures across tasks with opposite stimulus-response relationships were more aligned in simultaneously trained networks (Figure 8). `Pro` and `Anti` tasks shared a subspace for the memory-period stimulus representation. In contrast to mostly orthogonal subspaces in our sequentially trained networks (Figure 8A), simultaneously trained networks' memory-period activity was arranged such that trials with the same output had similar initial conditions during the go period (Figure 8B). Given this relationship between initial conditions, tasks with different stimulus-response relationships could share aligned go-period fixed-point structures (Figure 8C). We quantify the extent to which axes that capture the maximal stimulus variance are aligned during the go period in Figure 8D. These results verify that go-period activity across tasks with the opposite stimulus response is more aligned when networks are simultaneously trained compared to our sequentially trained networks.

Our learning rule restricts adjustments to dynamics within subspaces explored by previously learned tasks. This limits the network's ability to create trajectories that converge in state space after starting from different initial conditions. Simultaneously trained networks were able to produce convergent hidden-state trajectories into a shared subspace despite different stimulus-response relationships. This allowed the network to reuse output-related dynamical structures across all tasks. This shared organization could contribute to improved task performance in simultaneously trained networks (Figure 3A).

# 6 Discussion

We have introduced a novel approach for continual learning in RNNs to solve multiple tasks. Our approach is inspired by structure in cortical neural populations [2, 6–8] and encourages the network to organize task dynamics such that more (less) similar tasks evolve in aligned (orthogonal) subspaces.

We have shown that our approach significantly improved retention of previously learned tasks compared to weight change regularization-based approaches. The favorable performance of our method illustrates that continual learning in RNNs is better addressed by preventing interfering changes in the network's dynamics, rather than penalizing individual weight changes.

Our learning approach can also be viewed in terms of adaptive learning rates. Preconditioning methods [31–33] and FORCE learning [34] use within-batch correlational structure to adaptively change learning rates in order to speed up learning and convergence. In contrast, we use the correlational structure on previous tasks to *slow down* learning rates along specific directions in

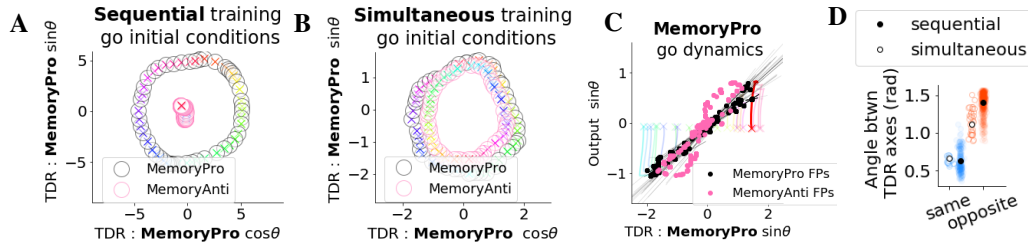

Figure 8: Simultaneously trained networks share output structure. Orientation of hidden state at start of go period (end of memory) colored by $\theta$ for **A:** sequentially **B:** simultaneously trained networks. **C:** Same as Figure 6C for simultaneously trained networks. **D:** Average of 2D subspace angle (rad) between network dimensions capturing maximum stimulus-related variance for tasks with same (blue) or opposite (orange) stimulus-response relationships during go period for sequentially and simultaneously trained networks. Means across all networks indicated in black.

weight space. Directions in parameter space that are well explored on previous tasks are modified more slowly, hinting towards a potential connection with uncertainty-based approaches towards adaptive learning rates [35].

We identified key differences between task organization in networks trained using our continual learning approach and those that arise in simultaneously trained networks. The effects of learning algorithms on network dynamics and learnt representations is an active area of research [see e.g. 36]. The results we presented here suggest that orthogonal representations may be favorable for retaining information that becomes inaccessible during later stages of learning, while other, more efficient, representations may arise when information about each task is accessible throughout learning. Our analyses reveal a potential explanation for the value of simultaneous review of related tasks or concepts (Figure 8). It will be interesting to investigate how our sequential training approach could be extended to better integrate dynamics across subspaces to share fixed point structures across more tasks. We could achieve this by including some form of consolidation of previously learned tasks using replay. Assessing whether neural population dynamics in humans or other animals may similarly be affected by different training paradigms is an interesting direction of future research.

We focus on toy examples so that we may analyze solutions the network obtains under different training regimes. We are interested in expanding this work to more complex applications, such as multi-task brain machine interface control and to more biologically plausible learning conditions without task boundaries.

## Broader Impact

This work proposes a novel continual learning algorithm which will contribute to the advance of related methods. Continual learning of dynamic tasks has not been well-explored in machine learning so far, but will likely be important for fields such as robotics and developing artificial intelligent agents more generally. Furthermore, we utilize the framework of recurrent networks to test and refine hypotheses about computation in biological systems. Advances in this area will contribute to the design of new experiments and aid the analyses of recorded data in the field of neuroscience.

## Acknowledgments and Disclosure of Funding

We would like to thank Jorge Menendez, Matt Golub, Ted Moskovitz and Benjamin Antin for helpful comments on the manuscript. This work was funded by the Simons Foundation (SCGB 543049, 543045, 543039; DS, KVS, MS) and the Gatsby Charitable Foundation.

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
