[Supplementary Material]

# Supplementary Material:
# Organizing recurrent network dynamics by task-computation to enable continual learning

**Lea Duncker**[*]
Gatsby Unit, UCL
London, UK
duncker@gatsby.ucl.ac.uk

**Laura N. Driscoll**[*]
Stanford University
Stanford, CA
lndrisco@stanford.edu

**Krishna V. Shenoy**
Stanford University
Stanford, CA
shenoy@stanford.edu

**Maneesh Sahani**[†]
Gatsby Unit, UCL
London, UK
maneesh@gatsby.ucl.ac.uk

**David Sussillo**[†]
Google Brain, Google Inc.
Mountain View, CA
sussillo@google.com

## Related work

We review a selection of previous approaches to continual learning that either apply a form of regularization to changes of the network weights, or modify the learning rule during training, since these are most relevant to the approach we introduce in section 4 of the main paper.

### Elastic Weight Consolidation

Elastic Weight Consolidation (EWC) was proposed in [1], and aims to slow down learning for weights that were deemed important for previous tasks. This is is achieved by adding an importance-weighted regularization term to the objective function, which ties a given parameter to its value after learning a previous task. The important weights are computed as the diagonal of the Fisher Information matrix $F$ evaluated at the parameter values $\theta_i^*$ at the end of training on the previous task. For a loss function $\mathcal{L}(\theta)$, the EWC objective is given by

$$\mathcal{L}(\theta) = \mathcal{L}(\theta) + \frac{\lambda}{2} \sum_i F_i (\theta_i - \theta_i^*)^2 \tag{1}$$

For the comparisons in Figure 3 of the main paper, the value of the regularization parameter $\lambda$ was determined by a coarse gridsearch and was set to $\lambda = 1e^5$ – the largest parameter for which the optimization remained numerically stable for all networks.

### Synaptic Intelligence

Synaptic intelligence was proposed in [2] and is another continual learning approach based on weight change regularization. The method aims to counteract catastrophic forgetting by trying to avoid drastic changes in important network parameters. Importance is measured via $\omega_i^\mu$ and reflects how much the connection weight $\theta_i$ has contributed to an improvement of the objective $\mathcal{L}(\theta)$ on task $k$. $\omega_i^\mu$ is computed online as the running sum of the product of the gradient of the loss with respect to the parameters and the change in parameters throughout training. The synaptic intelligence objective

---

[*]Equal Contribution
[†]Equal Contribution

Figure S1: Log-cost on test trials when training the first and second task under different modifications of stochastic gradient descent. One sided projections are less successful in retaining a low cost on the first task after training on the second task.

is given by

$$\mathcal{L}(\theta) = \mathcal{L}(\theta) + c \sum_i \Omega_i^k (\tilde{\theta}_i - \theta_i)^2, \quad \Omega_i^k = \sum_{\nu < k} \frac{\omega_i^\nu}{(\Delta_i^\nu)^2 + \xi} \tag{2}$$

where $\tilde{\theta}_i = \theta_i(t^{k-1})$ is the value of the weight at the end of training the previous task and $\Delta_i^\nu = \theta_i(t^\nu) - \theta_i(t^{\nu-1})$.

For the comparisons in Figure 3 of the main paper the regularization parameter $c$ was chosen by a coarse gridsearch and was set to $c = 1$ and $\xi = 0.01$. For larger values of $c$ the initial retention of tasks was improved, but the optimization eventually became unstable for all networks.

**Orthogonal Weights Modification**

The authors of [3] propose Orthogonal Weights Modification (OWM) as an approach for sequentially learning multiple classification tasks in feed-forward neural networks. The key idea is to allow for modifications of the network weights only along directions that are orthogonal to the subspace spanned by all previously encountered inputs. This is done by defining a projection matrix for task j

$$\mathbf{P}_\ell^j = \mathbf{I} - \mathbf{A}_\ell (\mathbf{A}_\ell^\mathsf{T} \mathbf{A}_\ell + \alpha \mathbf{I})^{-1} \mathbf{A}_\ell^\mathsf{T} \tag{3}$$

where $\mathbf{A}_\ell = [\mathbf{x}_1^\ell, \ldots \mathbf{x}_n^\ell]$ contains all previously encountered inputs to the network layer $\ell$ and $\alpha$ is a small constant used for regularization. The learning update is given by

$$\mathbf{W}^\ell \leftarrow \mathbf{W}^\ell - \kappa \mathbf{P}_\ell^j \nabla \mathbf{W}^\ell \tag{4}$$

where $\nabla \mathbf{W}^\ell$ is the standard gradient descent update obtained via backpropagation. In order to avoid having to store all input patterns encountered so far, the authors use a recursive least squares (RLS) approach [4] to computing $\mathbf{P}_\ell^j$.

Instead of projecting out previously encountered inputs only (as in OWM), our proposed learning rule modifies both sides of the gradient update. We compare modifications on either side of the gradient update to demonstrate that a double-sided modification reduces forgetting. In Figure S1, we repeat the experiments of Figure 3 using one-sided projections and plot the log-cost on test trials throughout training of the first and second task. The increase in cost after introducing the second task provides a comparison of the extent of forgetting across methods. Tasks are best retained using the double-sided approach because both input and output spaces can interfere across tasks unless we project out updates in these dimensions during learning.

## Supplementary methods

**Low-rank multi-task computation in RNNs**

Previous work in RNNs has shown that the low-dimensional solution to a cognitive task can be implemented in an RNN via a low-rank component in $\mathbf{W}^{\text{rec}}$ [5] and that individual tasks' low-rank components may be combined by summation to solve multiple tasks in the same network [6]. Similarly, a single dynamical system implementing multiple independent computations can be constructed via a dynamics matrix $\mathbf{W} = [\mathbf{W}^{\text{rec}} \ \mathbf{W}^{\text{in}}]$ of the form

$$\mathbf{W} = \sum_{k=1}^K \mathbf{W}_k = \sum_{k=1}^K \mathbf{U}_k \mathbf{V}_k^\mathsf{T} \tag{5}$$

where $\mathbf{W}_k$ is a low-rank component implementing a particular computation. We can think of the $\mathbf{W}_k$ independently, as long as the different low-rank dynamical components do not interfere with each other. To ensure this noninterference, we require $\mathbf{U}_k^\mathsf{T}\mathbf{V}_{k'} = 0$, for all $k \neq k'$, which translates into the output-space of one component being orthogonal to the input-space of another component. For general choices of $\phi(h)$, we also require that the non-linearity does not map the output-space $\mathbf{U}_k$ of one component into the input-space $\mathbf{V}_{k'}$ of another component, as this would cause the activity across dynamical components to interfere with each other.

**Continual learning algorithm details**

The algorithm we introduced in section 4 of the main paper relies on several projection matrices, which are used to modify the SGD learning update to enable continual learning. In order to compute the relevant projection matrices, we first compute the covariance matrices

$$\mathbf{\Sigma}_k^z = \sum_{t,r} \mathbf{z}_t^{k,r}\mathbf{z}_t^{k,r^\mathsf{T}} = \begin{bmatrix} \mathbf{\Sigma}_k^h & \mathbf{\Sigma}_k^{hx} \\ \mathbf{\Sigma}_k^{xh} & \mathbf{\Sigma}_k^x \end{bmatrix}, \qquad \mathbf{\Sigma}_k^y = \mathbf{Y}_{1:k}\mathbf{Y}_{1:k}^\mathsf{T} \tag{6}$$

from network activity generated on task $k$, averaged across time indices $t$ and trials $r$. The total covariance ($\mathbf{\Sigma}_{1:k}^z = \mathbf{Z}_{1:k}\mathbf{Z}_{1:k}^\mathsf{T}$, $\mathbf{\Sigma}_{1:k}^{wz} = \mathbf{W}\mathbf{Z}_{1:k}\mathbf{Z}_{1:k}^\mathsf{T}\mathbf{W}^\mathsf{T}$ and analogous for other covariances, averaging over data from all previously encountered tasks) is updated online after each task via

$$\mathbf{\Sigma}_{1:k}^* = \frac{1}{k}\sum_{\ell=1}^{k}\mathbf{\Sigma}_\ell^* = \frac{k-1}{k}\frac{1}{k-1}\sum_{\ell=1}^{k-1}\mathbf{\Sigma}_\ell^* + \frac{1}{k}\mathbf{\Sigma}_k^* = \frac{k-1}{k}\mathbf{\Sigma}_{1:k-1}^* + \frac{1}{k}\mathbf{\Sigma}_k^* \tag{7}$$

where $* = \{wz, z, y, h\}$.

The projection matrices we use for our continual learning algorithm are computed as

$$\mathbf{P}_{wz}^{1:k} = \left(\alpha^{-1}\mathbf{W}\mathbf{\Sigma}_{1:k}^z\mathbf{W}^\mathsf{T} + \mathbf{I}\right)^{-1} \tag{8}$$

$$\mathbf{P}_z^{1:k} = \left(\alpha^{-1}\mathbf{\Sigma}_{1:k}^z + \mathbf{I}\right)^{-1} \tag{9}$$

$$\mathbf{P}_h^{1:k} = \left(\alpha^{-1}\mathbf{\Sigma}_{1:k}^h + \mathbf{I}\right)^{-1} \tag{10}$$

$$\mathbf{P}_y^{1:k} = \left(\alpha^{-1}\mathbf{\Sigma}_{1:k}^y + \mathbf{I}\right)^{-1} \tag{11}$$

Our proposed continual learning algorithm is summarized in Algorithm 1.

In the main paper, we have motivated our algorithm in terms of preserving the input/output space of the dynamics on previous tasks by projecting away interfering dimensions from the weight update. This is effectively implementing a solution of the same form as Equation (5), where the orthogonal projections ensure that a new low-rank component required to solve the new task – added to the system via the gradient update – will not interfere with existing dynamics. In this way, the projection matrices are exactly implementing the constraints on the input/output space mentioned above in the context of $\mathbf{U}_k$ and $\mathbf{V}_k$.

An alternative interpretation of the action of the projection matrices is in terms of slowing down the learning rate along previously explored directions in network-activity space. Our algorithm implements an adaptive learning rate schedule across tasks, where directions of fast or slow learning depend on the dynamical input/output space of previous tasks. This interpretation becomes more apparent when explicitly considering the action of the projection matrices on the gradient. Letting $\mathbf{P}_*^{1:k} = \mathbf{U}_*^{1:k}\mathbf{\Gamma}_*^{1:k}\mathbf{U}_*^{1:k^\mathsf{T}}$ denote the eigendecomposition of the projection matrices, we can express the weight update as

$$\Delta\mathbf{W}_{\mathsf{CL}} = \mathbf{P}_{wz}^{1:k}\,\nabla_\mathbf{W}\mathcal{L}\,\mathbf{P}_z^{1:k} = \mathbf{U}_{wz}^{1:k}\mathbf{\Gamma}_{wz}^{1:k}\mathbf{U}_{wz}^{1:k^\mathsf{T}}\,\nabla_\mathbf{W}\mathcal{L}\,\mathbf{U}_z^{1:k}\mathbf{\Gamma}_z^{1:k}\mathbf{U}_z^{1:k^\mathsf{T}} \tag{12}$$

$$\mathbf{U}_{wz}^{1:k^\mathsf{T}}\Delta\mathbf{W}_{\mathsf{CL}}\mathbf{U}_z^{1:k} = \mathbf{\Gamma}_{wz}^{1:k}\mathbf{U}_{wz}^{1:k^\mathsf{T}}\,\nabla_\mathbf{W}\mathcal{L}\,\mathbf{U}_z^{1:k}\mathbf{\Gamma}_z^{1:k} \tag{13}$$

$$\Delta\widetilde{\mathbf{W}}_{\mathsf{CL}} = \mathbf{\Gamma}_{wz}^{1:k}\,\widetilde{\nabla_\mathbf{W}}\mathcal{L}\,\mathbf{\Gamma}_z^{1:k} \tag{14}$$

where $\Delta\widetilde{\mathbf{W}}_{\mathsf{CL}}$ denotes the weight update rotated into the relevant input/output space as defined by the eigenbasis of the projection matrices. The weight update to the $ij$-th entry in the weight matrix

can be expressed in this rotated space as

$$[\Delta\widetilde{\mathbf{W}}_{\text{CL}}]_{ij} = \frac{\alpha^2[\nabla_{\widetilde{\mathbf{W}}}\mathcal{L}]_{ij}}{(\lambda_i^{wz} + \alpha)(\lambda_j^z + \alpha)} \tag{15}$$

where $\lambda_i^*$ denotes the $i$th eigenvalue of $\Sigma_{1:k}^*$, for $* = \{wz, z, y, h\}$. This shows that the learning rate of the weight updates is scaled by the inverse eigenvalue of the covariance of network activity on previous tasks. Our learning algorithm hence implements an adaptive learning rate schedule dependent on the total variance of activity along input/output directions on previous tasks. If the network produces a lot of variance along a given direction during a previous tasks, the total variance along this direction $\lambda_i^*$ will take on a larger value and therefore slow the learning rate along the associated direction on later tasks. Crucially, this slowing depends on interactions of the total amount of variance in the input and output space of the dynamical system, as is apparent from Equation (15).

---

**Algorithm 1:** sequential training via orthogonal task dynamics

---

**Input:** task sequence $\mathcal{T}_1, \mathcal{T}_2, \ldots \mathcal{T}_K$, learning rate $\eta$, maxiter
**Output:** $\mathbf{W}, \mathbf{W}^{\text{rec}}$
initialization;
$\mathbf{P}_{wz}^0 = \mathbf{I}, \ \mathbf{P}_z^0 = \mathbf{I} \ \mathbf{P}_h^0 = \mathbf{I} \ \mathbf{P}_y^0 = \mathbf{I}$;
**for** *task* $k = 1, \ldots K$ **do**
    **for** *iter* $i = 1, \ldots$ *maxiter* **do**
        $\mathbf{W} \leftarrow \mathbf{W} - \eta\mathbf{P}_{wz}^{1:k-1} \nabla_{\mathbf{W}}\mathcal{L} \ \mathbf{P}_z^{1:k-1}$ ;
        $\mathbf{W}^{\text{out}} \leftarrow \mathbf{W}^{\text{out}} - \eta\mathbf{P}_y^{1:k-1}\nabla_{\mathbf{W}^{\text{out}}}\mathcal{L} \ \mathbf{P}_h^{1:k-1}$ ;
    **end**
    $\mathbf{\Sigma}_k^z, \mathbf{\Sigma}_k^y \leftarrow$ covariance on trials from task $k$;
    $\mathbf{P}_{wz}^{1:k}, \mathbf{P}_z^{1:k}, \mathbf{P}_h^{1:k}, \mathbf{P}_y^{1:k} \leftarrow$ update projection matrices;
**end**

---

### Network training details

Networks were trained to minimize the squared error between the output activations and the targets, with added $L_2$-norm weight (regularization weight parameter $1e^{-5}$) and activity regularization(regularization weight parameter $1e^{-7}$). The regularization parameters were determined through a gridsearch. We used 64 trials per minibatch during training. $maxiter$ was set to $1.25e^7/64$ per task. In practice we used SGD with momentum $\nu = 0.9$, still applying the projection matrices to the gradient. The learning rate was set to $\eta = 0.001$. The covariance is computed on validation trials for each tasks. We set $\alpha = 0.001$ for all networks that were trained using our continual learning algorithm.

### Task performance measure

The decoded response direction at the last time step of the trial is considered correct if it is within 10% of the target direction (within $\frac{2\pi}{10}$). If the activity of the fixation output falls below 0.5, the network is considered to have broken fixation. Average performance was calculated across 20 iterations for 1000 trials for each task.

### Targeted Dimensionality Reduction (TDR)

We closely follow the method originally outlined in (Mante, Sussillo et al. 2013) [7]. We used multi-variable, linear regression to determine how stimulus inputs affect the responses of each hidden unit at a given time in the trial. We describe the hidden unit responses of unit $i$ as a linear combination of stimulus inputs:

$$\mathbf{h}_i(k) = \boldsymbol{\beta}_i(0)\cos\theta_k + \boldsymbol{\beta}_i(1)\sin\theta_k + \boldsymbol{\beta}_i(2) \tag{16}$$

where $\mathbf{h}_i(k)$ is the response of unit $i$ at a given timestep on trial $k$ and $\theta_k$ is the stimulus angle on trial $k$. The regression coefficients $\boldsymbol{\beta}_i(v)$ for $v = 0, 1$ describe how much the trial-by-trial activity of unit $i$, at a given time in the trial, depends on $\cos\theta_k, \sin\theta_k$ respectively. $\boldsymbol{\beta}_i(2)$ is a bias term.

To estimate the regression coefficients $\boldsymbol{\beta}_i(v)$, we define for each unit a matrix $\mathbf{F}_i$ of size $N_{\text{coef}} \times N_{\text{trial}}$, where $N_{\text{coef}} = 3$ is the number of regression coefficients to be estimated, and $N_{\text{trial}} = 400$ is the number of trials used in the regression analysis. The regression coefficients can be expressed as:

$$\boldsymbol{\beta}_i = (\mathbf{F}_i \mathbf{F}_i^\mathsf{T})^{-1} \mathbf{F}_i \mathbf{h}_i \tag{17}$$

where $\boldsymbol{\beta}_i$ is a vector of length $N_{\text{coef}}$ with elements $\boldsymbol{\beta}_i(v)$, $v = 0, 1, 2$. We rearrange entries of $\boldsymbol{\beta}_i$ of length $N_{\text{coef}}$, into a new set of vectors, $\boldsymbol{\beta}_v$ of length $N_{\text{units}}$. These new vectors correspond to directions in state space that capture maximal variance of the input variables $v$. We orthogonalize regression vectors with the QR-decomposition:

$$\boldsymbol{B} = \boldsymbol{Q}\boldsymbol{R} \tag{18}$$

where $\boldsymbol{B} = [\boldsymbol{\beta}_0, \boldsymbol{\beta}_1, \boldsymbol{\beta}_2]$ is a matrix whose columns correspond to regression vectors $\boldsymbol{\beta}_v$. $\boldsymbol{Q}$ is an orthogonal matrix and $\boldsymbol{R}$ is an upper triangular matrix. The first two columns of $\boldsymbol{Q}$ correspond to the orthogonalized regression vectors $\boldsymbol{\beta}_v^\perp$, which we refer to as TDR axes related to $\cos\theta$ and $\sin\theta$ inputs respectively. TDR axes were defined separately for each task producing an independent set of TDR axes related to input variables for each task, $\boldsymbol{\beta}_v^{\mathsf{TaskA}}$. Axes were typically defined by responses on the last timestep of a task period of interest, with the exception of Figure 8A,B where initial conditions are visualized in a subspace defined by the first timestep of the go period.

We note that a key difference between our approach and the original implementation of TDR is that we skip the de-noising step. Because we only applied this method to simulated data, we could simply remove the added shared and private noise in the inputs and hidden units respectively.

**Calculating subspace angles**

We calculate the angle between TDR axes identified during stimulus and go periods for sequentially and simultaneously trained networks.

$$\theta_v = \cos^{-1} \frac{\boldsymbol{\beta}_v^{\mathsf{TaskA}} \cdot \boldsymbol{\beta}_v^{\mathsf{TaskB}}}{\|\boldsymbol{\beta}_v^{\mathsf{TaskA}}\| \|\boldsymbol{\beta}_v^{\mathsf{TaskB}}\|} \tag{19}$$

We are finding the angles $\theta_v$ between TDR axes $\boldsymbol{\beta}_v^{\mathsf{TaskA}}$ and $\boldsymbol{\beta}_v^{\mathsf{TaskB}}$ for $v = 0, 1$. Angles are computed for comparisons between axes w.r.t. the same input (either $\cos\theta$ or $\sin\theta$) across different tasks within the same network.

**Finding fixed points in trained networks**

We studied RNN dynamics by reducing their nonlinear dynamics to linear approximations. We first write down the update equation.

$$\mathbf{h}_{t+1} = \phi(\mathbf{W}^{\mathsf{rec}}\mathbf{h}_t + \mathbf{W}^{\mathsf{in}}\mathbf{x}_t + \boldsymbol{\xi}_t) = F(\mathbf{h}_t, \mathbf{x}_t) \tag{20}$$

We optimize to find fixed points $\{\mathbf{h}_1^*, \mathbf{h}_2^*, ...\}$ of an RNN such that $\mathbf{h}_i^* \approx F(\mathbf{h}_i^*, \mathbf{x}^*)$. These positions in state space are called fixed points because they remain in the same location on each update step. We use the term fixed point to include approximate fixed points, which are slow enough on the timescale of our tasks that they appear to be fixed.

We identify fixed points during each task period separately. Each task period is defined by a temporal segment where task inputs are static ($\mathbf{x}^*$). These static inputs are given to the system during fixed point finding. Numerical procedures for identifying fixed points are described in [8, 9]. Around each fixed point, the local behavior of the nonlinear system can be approximated as a linear one:

$$\mathbf{h}_{t+1} \approx \mathbf{h}^* + \mathbf{J}(\mathbf{h}^*, \mathbf{x}^*)(\mathbf{h_t} - \mathbf{h}^*) \tag{21}$$

where $\mathbf{J}_{ij}(\mathbf{h}^*, \mathbf{x}^*) = \frac{\partial F_i(\mathbf{h}^*, \mathbf{x}^*)}{\partial h_j^*}$ denotes the Jacobian of the RNN update rule. We studied these linear systems using eigenvector decomposition for non-normal matrices as in [8–10]:

$$\mathbf{J} = \mathbf{R}\boldsymbol{\Lambda}\mathbf{L} = \sum_{a=1}^{N} \lambda_a \mathbf{r}_a \mathbf{l}_a^\mathsf{T}, \tag{22}$$

Figure S2: The log cost for each task on 5 example networks trained using our continual learning approach. The cost is computed on test trials under different learning schedules for each panel. The color shading indicates which task was trained on each epoch of training.

where the columns of $\mathbf{R}$ $(\mathbf{r}_a)$ and the rows of $\mathbf{L}$ $(\mathbf{l}_a^\mathsf{T})$ are the right and left eigenvectors of $\mathbf{J}$ respectively. $\mathbf{\Lambda}$ is a diagonal matrix of the associated eigenvalues, $\lambda_a$. For some task periods we studied, there was a single fixed point and all $\lambda_a < 1$, suggesting dynamics were contracting towards a single fixed point in all dimensions. In other task periods, we found a collection of fixed points where some $\lambda_a > 1$. We examined all eigenvectors $(\mathbf{r}_a)$ associated with slightly expanding dimensions to better understand their relationship with hidden unit activity. To do this, we projected right eigenvectors into axes identified through targeted dimensionality reduction (TDR) or using vectors of $\mathbf{W}^{\text{out}}$ (Output). These projections revealed collections of fixed points together formed ring attractors (both in memory and go periods of `Memory` tasks).

## Supplementary results

### Effect of training order

Figure S3: Effects of task order and network size. **A**: Final log-cost at the end of training on the respective tasks for tasks that were trained 1st, 2nd, 3rd or 4th. **B**: Final log-cost on `DelayPro`, trained 1st for different network sizes

We show the behavior of the log loss on test trials throughout training for example task orderings in Figure S2. We found that our continual learning approach is mostly robust to task orderings during training. The performance was affected only for training orders where the `MemoryPro(Anti)` task was trained first, followed by `DelayPro(Anti)`, where retention of the `DelayPro(Anti)` task appeared to be worse (Figure S2 middle column).

Tasks that were trained first generally achieved the lowest test error (Figure S3A), consistent with the first task being the least constrained in terms of the size and orientation of the subspace the dynamics can explore. In support of this view, we found that single tasks achieved slightly lower test errors in larger networks (Figure S3B).

### Effect of network size

Figure S3B shows some effect of network size on training a single task. We next tested whether our continual learning approach was sensitive to the number of recurrent units in the network. Figure **??**

Figure S4: The log cost for each task on 5 example networks trained using our continual learning approach under different numbers of recurrent units. The color shading indicates which task was trained on each epoch of training.

Figure S5: Same as Figure S4 for networks with 256 recurrent units and different choices of activation function.

shows the log cost on test trials for each task for three network sizes. Networks with fewer recurrent units had a lower capacity, which made separating task components into orthogonal subspaces harder. We found that the performance of our algorithm was worse for networks with 64 recurrent units, but comparable for networks with 128 units.

**Effect of activation function**

Our continual learning approach is based on intuition from linear systems, and all results of the main paper were based on networks with rectified linear (ReLU) activation functions. Figure S5 shows the log-cost on test trials throughout sequential training under other choices of nonlinear activation function. Notably, the retention of the first task was worse, while the retention of later tasks was comparable. Compared to the log-losses in Figure 3D&E of the main paper, our algorithm is competitive even under more general choices of activation function. However, the extension of our algorithm to highly nonlinear settings, where projections based on linear subspaces will be less effective is a direction of future work.

**Performance on decision-making task**

We have focused on a small set of tasks in the main paper. To show that our learning approach is also effective on other tasks, we include the test error throughout training on a set of perceptual decision-making tasks in Figure S6. In these tasks, networks receive two stimulus inputs that are separated in time by a delay period. A decision has to be made via a response along the direction of the stronger stimulus. Inputs are structured as before, but now involve two modalities. The network either receives input for only one of the two modalities (`MemoryDM` tasks), or both. When the network receives inputs for both modalities, it has to learn to contextually ignore input for one of the modalities (`ContextMemoryDM` tasks), or respond in the direction of the stimulus with highest combined strength across both modalities (`MultiMemoryDM` task). Further details may be found in [11].

**Fixed point structure for `Anti` tasks**

Fixed point structures across `Pro/Anti` tasks were similar, but with rotated relationships to output dimensions. Figure S7 shows the `Anti`-task analogue to Figure 5 of the main text.

Figure S6: Log cost for each tasks on 5 example networks (with different random seeds) trained under different orderings of decision-making tasks.

Figure S7: As in Figure 5, fixed points (black) and hidden state activity (red) for a single trial, $\theta = \frac{\pi}{2}$. Activity from other trials of the same task are faded (colored by $\theta$) emanating from 'x'. Axes identified through targeted dimensionality reduction (TDR) or using vectors of $\mathbf{W}^{\text{out}}$ (Output). Insets are overlaid eigenspectra of all period specific fixed points. **A-B:** `DelayAnti` stimulus, go period. **C:** `MemoryAnti` stimulus period. **D-E:** `MemoryAnti` memory period ring attractor in two subspaces. **F-G:** `MemoryAnti` go period ring attractor in two subspaces. **D-G:** Unstable right eigenvectors of linearized dynamics around each fixed point are projected into activity subspace (black lines).

**Proof of descent direction**

We can show that the direction of the remaining update is still a valid descent direction on the loss. Letting $u = -\mathbf{P}_1 \nabla_{\mathbf{W}} \mathcal{L} \mathbf{P}_2$ and $g = \nabla_{\mathbf{W}} \mathcal{L}$, we require that $\langle u, g \rangle \leq 0$ for $u$ to be a descent direction on the loss $\mathcal{L}$.

$$\langle u, g \rangle = -\text{Tr}\left[\nabla_{\mathbf{W}}\mathcal{L}^{\mathsf{T}} \mathbf{P}_1 \nabla_{\mathbf{W}}\mathcal{L} \mathbf{P}_2\right] = -\text{vec}(\nabla_{\mathbf{W}}\mathcal{L})^{\mathsf{T}}(\mathbf{P}_2 \otimes \mathbf{P}_1)\text{vec}(\nabla_w \mathcal{L}) \leq 0 \qquad (23)$$

where $\text{vec}(\cdot)$ denotes the vectorization operation. It holds that $\langle u, g \rangle \leq 0$ when $\mathbf{P}_1$ and $\mathbf{P}_2$ are positive semidefinite matrices – a property that is conserved under the Kronecker product.