[Reviews · NeurIPS 2020]

Review 1

Summary and Contributions: This manuscript addresses the problem of continual learning in RNN. The authors propose a new learning rule that allows to organize the dynamics for different tasks into orthogonal subspaces. Using a set of neuroscience tasks, they show how this learning rule allows to avoid catastrophic interferences between tasks. By analyzing the dynamics of trained networks they provide evidence for why their learning rule is successful, it also allows them to discuss the problem of transfer learning.

Strengths: - propose a new original solution to the problem of continual learning, which also allows them to address and understand under which conditions learning in one task can be transfered to learning off another task. This should be of interest to the NeurIPS community.

Weaknesses: - limited to a subset of tasks on which RNN are used.

Correctness: Claims, method and methodology looks correct

Clarity: The paper is well written

Relation to Prior Work: yes

Reproducibility: Yes

Additional Feedback: Update: the authors response shows that the clarity of the manuscript will improve and that the relationship with other works will be made more exhaustive. I thus still think this paper should be presented at NeurIPS and keep my score at 8. ------------ I liked this work very much. It proposes a new learning rule to solve an important problem. Comparisons with previous approaches are very welcome, and the insights gained by analyzing the network dynamics allows to clearly understand why and how this learning rule works.


Review 2

Summary and Contributions: The authors propose a modification to SGD for sequential learning of multiple tasks. The algorithm is very similar to online learning (FORCE / RLS), but at the task level and not the timestep level. The algorithm slows down learning in the directions in phase space visited by previous tasks, thereby reducing interference. The authors show that interference is reduced compared to other recent algorithms. Furthermore, learning a new task is accelerated by previously learning a related one. The authors then perform a dynamical-system analysis, elucidating the fixed point structure of the tasks. The analysis reveals that dynamical structures are reused, possibly contributing to the faster learning observed. Furthermore, the authors show that the dynamical structure of simultaneously trained tasks is qualitatively different from that generated by sequential learning of the same tasks.

Strengths: 1. The performance of the model is better than recent previous proposals for avoiding catastrophic forgetting. 2. The saving effect when sequentially learning related tasks is similar to transfer learning, and I don’t think it has been studied in this manner before. 3. The dynamical systems insights are both interesting and novel.

Weaknesses: 1. Fixed point structure was only shown for a single task or after two tasks. The modification of the structure created by the first task as a result of learning the second task was not presented or discussed. 2. Part of the motivation for the paper is biological. As such, there is no discussion about plausibility of the learning rule. 3. The algorithm requires some “bookkeeping”. Task boundaries need to be well defined. Activity correlation has to be accumulated across tasks.

Correctness: The claims and methods appear to be correct.

Clarity: The paper is clearly written. The figures are rather small.

Relation to Prior Work: The authors relate to prior work regarding catastrophic forgetting, but don’t relate to recursive least squares and FORCE learning. It seems to me that the proposed algorithm is highly similar to FORCE learning (Sussillo & Abbott, Neuron 2009) but at the level of tasks instead of timesteps. The matrix inversion is also done explicitly in this case, instead of incrementally updating the inverse matrix. This is probably because inversion only happens once for every task. Related to this FORCE similarity is the work by Beer & Barak (Neural Computation, 2019) where catastrophic forgetting and its alleviation are studied in RNNs within a single task.

Reproducibility: Yes

Additional Feedback: POST REBUTTAL UPDATE: After reading the other reviews and rebuttal, I raise my score to an 8.


Review 3

Summary and Contributions: [The problem they focus on] This paper works on the multi-task continual learning problem, which remains a challenging problem in machine learning due to catastrophic interference between current and previous tasks. The authors study the problem from the perspective of dynamical systems which are popular analysis tool in neural data, e.g., spike train, EEG, or other time series data. [Basic assumption in the paper] The author has assumed some subspace structures existing during sequential tasks learning from some recent experimental results from neural population. They assume that either orthogonal or shared subspace provide a elegant organization of the neural dynamics, which leads to a robust and efficient computations for multi-tasks. Briefly, similar tasks could evolve under similar dynamics in a shared subspace, while dissimilar tasks (or dissimilar components of tasks), could be confined to orthogonal subspaces. [Their solution or contributions] They claim to propose a novel continual learning algorithm that encourages networks to organize dissimilar dynamics into orthogonal subspaces, which I have proposed my concern in the next comments. [Why it works] The method can try to limit the interference for unrelated tasks or task components. Thus, the intrinsic space can learn the task independently. [The results] The paper shows that their proposed learning algorithm outperforms previous regularization-based continual learning approaches on four tasks which are used for studying the mutli-task representations in RNNs. The findings are also interesting but not very novel that they find shared structure across similar tasks facilitates faster learning during sequential training. Also, they highlight key differences w.r.t the task alignments for networks trained under sequential approach compared to simultaneously trained on all tasks. I think the key results are verifying that the Continual Learning in RNNs also help provide insights in the dynamics across multi-task learning.

Strengths: [Theoretical grounding] The proposed method seems effective. The core mathematical analysis is from equation (3) (4) (5) for updating the weights (making the orthogonal task dynamics possible). I have also gone through Supplementary Material, and find no novel derivations (correct me if you indicate any novel theoretical findings in the work). EWC, OWM, SI, low-rank multitask learning are also well-studied in previous works. [Empirical evaluation] Experiments of this paper are credible. I like the experiments done in this work. In summary, it clearly shows the task performance (via log cost on test trials) of networks trained with simultaneous, single task, fine tuning using SGD, EWC, SI, and the proposed method; the training order is also analyzed (although with very little effort discussing it); It is very clear for me to get the shared and independent latent/hidden dynamics across the experiments. [Novelty of the contribution] I think the main innovation is applying the OWM method into RNN setup, and then propose to solve neuroscience tasks. [Relevance to the NeurIPS community] I agree that the paper is quite relevant to the community.

Weaknesses: Although I admit that the proposed method works well on the sequential tasks. I have following main concerns and some of the weaknesses: a. It seems that the paper only applies the OWM[1] idea into RNNs condition. As I go through the two paper (OWN and this one) The two main differences are that: OWM focuses on feed-forward network (OWM works on few-shot scenario image classification problem) while the proposed method focus on RNNs. However, I would say from the Continual learning perspective, it is easy and straightforward to extend the orthogonal project in OWM:$P = I - Z ( Z^{T} Z + αI ) Z^{T}$ into RNNs condition. The propagation of OWM is: $Y = W Z$ while the propagation in this paper on RNN case is: $h^{t+1} = W Z$ (replacing the original output with the hidden state $t$) and $y^{t+1} = W_{out} h^{t+1} = W_{out} (WZ)$, which can be analogy as make OWM propagation twice. Thus, the orthogonal project of (3), (4) can be involuntary got. Correct me if I lost some of your key derivations.
 The OWM uses a “single-side update strategy” for weight updates, which only concerns the orthogonal project of input feature P_{in}. While the proposed method uses a “double-side update” which can concern both input project P_{in}(P_{wz} for W, P_{y} for W_{out}) and output project (P_{z} for W, P_{h} for W_{out}). Due to the fact that input and output both reflect the impact of the learning weight, I think this modification seems convincing and effective. However, it’s still essential to given some experiments to prove this, for example adding the additional experiments with update strategy: $△W = P_{wz}(▽L), △W_{out} = P_{y}(▽L)$ or $△W = (▽L) P_{y}, △W_{out} = (▽L) P_{h}$. b. The related work and the baseline methods seem not complete and the not state-of-the-art methods in Continual Learning field. The author claims “Previous work on continual learning in neural networks has mostly focused on feedforward architecture”. In the following works, Li et al (ICLR 2020) [2] propose a new scenario of continual learning which handles sequence-to-sequence tasks common in language learning. Ororbia et al (TNNLS) [3] propose the Parallel Temporal Neural Coding Network (P-TNCN), a biologically inspired model trained by the learning algorithm, which also focuses on RNN training. Cossu et al (IJCNN 2020) [4] proposes a RNN model for CL that is able to deal with concept drift in input distribution without forgetting previously acquired knowledge. I just list some of them since more work can be discussed for comparison (even the comparisons are not fair, but we had better discuss them to let the audience know the CL + RNN based solutions). c. The EWC and SI regularization based methods are one of the strategy, others including replay based and dynamic architecture based are also candidates. For most of the regularization based CL methods, the performances are not as good as the others. Can the authors discuss about it briefly and why not compare with other strategies? d. Most of experiments of this paper focuses on simulation datasets which is a bit simple. I’m also interested about the effectiveness on some real-world tasks used in RNNs, such as action classification or action anticipation on Kinetics. Can the author claim some of real-world usages that we need continual learning within RNN to develop the solutions ? [1] Guanxiong Zeng, Yang Chen, Bo Cui, and Shan Yu. Continual learning of context-dependent processing in neural networks. Nature Machine Intelligence, 1(8):364–372, 2019. [2] Yuanpeng Li, Liang Zhao, Kenneth Church, Mohamed Elhoseiny. Compositional language continual learning. ICLR 2020 [3] A. Ororbia, A. Mali, C. L. Giles and D. Kifer, Continual learning of recurrent neural networks by locally aligning distributed representations, in IEEE Transactions on Neural Networks and Learning Systems. [4] Andrea Cossu, Antonio Carta, Davide Bacciu, Continual learning with gated incremental memories for sequential data processing. IJCNN 2020

Correctness:  The claims and methods are correct in this paper.

Clarity: a. This paper is well written and easy to follow. Some minor problems: Line 43: Shared structure -> The shared structure; Line 70: feed back —> do you mean the “is feed into” ; Line 80: structure -> the structure; Line 89 & 120: along -> along with; Line 133: and go -> , and go; Line 138: for an example -> for example (sorry that the first term is not very widely used in writing); Line 161: this suggest -> this suggests; Line 171: stimulus dependent -> stimulus-dependent; Line 173: continuous -> The continuous; Line 179: information -> the information; Fig 6 caption: Each dot represents angle -> Each dot represents the angle; Fig8 caption: simultaneously trained network -> simultaneously trained networks. b. In the references, some of them have been published or accepted, better not use arXiv version, e.g., in the paper, reference [15] AISTATS 2020; reference [25] NeurIPS 2019 Workshop Neuro AI. c. Please also consistent with journal/conference names (abbreviation or full name).

Relation to Prior Work: The work is highly motivated by reference [1] OWM work. The details have been explained in above section. The RNN that focuses on continual learning or lifelong learning (avoid catastrophic problem) needs more relevant work comparisons, e.g., [2],[3],[4]. [1] Guanxiong Zeng, Yang Chen, Bo Cui, and Shan Yu. Continual learning of context-dependent processing in neural networks. Nature Machine Intelligence, 1(8):364–372, 2019. [2] Yuanpeng Li, Liang Zhao, Kenneth Church, Mohamed Elhoseiny. Compositional language continual learning. ICLR 2020 [3] A. Ororbia, A. Mali, C. L. Giles and D. Kifer, Continual learning of recurrent neural networks by locally aligning distributed representations, in IEEE Transactions on Neural Networks and Learning Systems. [4] Andrea Cossu, Antonio Carta, Davide Bacciu, Continual learning with gated incremental memories for sequential data processing. IJCNN 2020

Reproducibility: Yes

Additional Feedback: The overall writing and results in the work is quite good, and I am positive about the paper, if the author can claim more clearly about their contributions, I may raise my score. The main concern is the innovation part. _____________________________________ I have gone the response letter, and appreciated the authors feedbacks. For me,the updated rule is very similar with owm


Review 4

Summary and Contributions: This paper suggests an approach to train recurrent neural networks sequentially on several different tasks such that the interference between tasks is reduced and the network could retain the previously learned skills. The approach forces the network to learn each new task in its own orthogonal subspace by explicitly orthogonalizing the gradient descent updates. The approach is inspired by orthogonal subspaces of neural activity found in behaving animals in computational neuroscience.

Strengths: Disclaimer: I am not a specialist in training recurrent neural networks and cannot judge on the novelty of this paper or on the importance of this reseach within the RNN field. However, it seems that the paper addresses a meaningful problem, suggests an elegant (albeit very simple) solution, and has nice neuroscientific motivation.

Weaknesses: That said, I am concerned by some of the presented experimental evidence. I might have misunderstood some of it, so I could change my overall evaluation if the authors clarify these points. This concerns the claim that the networks exhibits transfer learning between related tasks (section 5.2) and the claim that the network uses shared representations for related tasks (section 5.4). See below.

Correctness: Yes

Clarity: Yes

Relation to Prior Work: Yes

Reproducibility: No

Additional Feedback: This paper suggests an approach to train recurrent neural networks sequentially on several different tasks such that the interference between tasks is reduced and the network could retain the previously learned skills. The approach forces the network to learn each new task in its own orthogonal subspace by explicitly orthogonalizing the gradient descent updates. The approach is inspired by orthogonal subspaces of neural activity found in behaving animals in computational neuroscience. Disclaimer: I am not a specialist in training recurrent neural networks and cannot judge on the novelty of this paper or on the importance of this reseach within the RNN field. However, it seems that the paper addresses a meaningful problem, suggests an elegant (albeit very simple) solution, and has nice neuroscientific motivation. That said, I am concerned by some of the presented experimental evidence. I might have misunderstood some of it, so I could change my overall evaluation if the authors clarify these points. This concerns the claim that the networks exhibits transfer learning between related tasks (section 5.2) and the claim that the network uses shared representations for related tasks (section 5.4). See below. Major comments * Section 5.2 claims transfer learning, i.e. faster learning of a task similar to the one learned before. The evidence (Figure 4) seems to be that if network learns first task A and then task B, then task B is learnt faster when A (A1) was similar to B, compared to the situtation when A (A2) was not similar to B. I have two issues here: a) Wouldn't one want to compare the speed of learning B with learning B on its own, without learning any A first? Transfer learning means that B is learnt faster after A1 compared to when it is learnt in isolation. b) It is not entirely clear to me if the ortoghonality was imposed while learning B in the same way independent of task A (i.e. for both A1 and A2). In case orthogonality was only imposed when learning B after A2 but not after A1, the results would become trivial. * Section 5.4 claims that similar tasks occupy shared subspaces. Here I have the same issue as (b) above: was the gradient descent orthogonality equally imposed when learning all these tasks? Some of the formulations in this section are very ambiguous/confusing: line 195 "[similar tasks] were allowed to evolve in the same subspace because their dynamics did not interfere" -- what does "were allowed" mean? If it means that orthogonality was not imposed during gradient descent, then the result is trivial. If it means that the orthogonality was imposed but network somehow still used the same subspace, then the result is nontrivial and interesting. * In the latter case, I don't quite understand how the subspaces can end up being shared if orthogonality between them was imposed during gradient descent. Medium comments * line 110: I did not understand why the orthogonalizatio is carried out in both the row and column spaces. I would intuitively think that one orthogonalization should be enough to keep the activity in orthogonal subspaces. E.g. in neuroscience if two patterns of activity are said to occupy orthogonal subspaces, this means orthogonality in the space of neurons, i.e. columns of X (and not in the space of time-points, i.e. rows of X). It seems it's motivated in line 120, but I did not understand that motivation either. * line 119: maybe give explicit formulas for the readouts here too. Minor comments * Figure 3B title -- "trianing" * line 25: You might want to cite some work by the Machens lab here, such as https://www.jneurosci.org/content/30/1/350 and https://elifesciences.org/articles/10989. (Disclaimer: I am not Christian Machens). There might be other relevant literature in neuroscience too. ------------------ POST-RESPONSE UPDATE: I said that I would be willing to revise my score if the authors clarified some important experimental details, so I am changing it to 6. That said, the authors' response did not really clarify my main confusion. E.g. section 5.4 claims that similar tasks end up occupying shared subspaces. How is this possible if the learning algorithm explicitly enforces orthogonality between subspaces used by different tasks? I hope this is clarified in the final version.

[Author Response · NeurIPS 2020]

We thank the reviewers for their careful reading, feedback and helpful comments and address specific concerns below.

**Novelty of contributions.** The novelty of our contribution is
two-fold: First, our proposed learning rule with modifications
to both sides of the gradient update is novel. This feature also
distinguishes our method from OWM (**R3**) and FORCE (**R2**).
Following **R3**'s suggestions, we repeat the experiments of Figure
3 using one-sided projections and plot the log-cost on test trials

throughout training of the 1st and 2nd task. The increase in cost after introducing the second task provides a comparison
of the extent of forgetting across methods. Tasks are best retained using the double-sided approach. **R4** raised that
they did not understand the motivation for the double-sided learning rule. In brief, both input and output spaces can
interfere across tasks unless we project out updates in these dimensions during learning. We hope the additional results
are convincing and we will further expand on the motivation offered in lines 107-111 when we revise the text. Second,
a key contribution of this work is the dynamical systems analysis of how our learning algorithm shapes the organization
of multiple tasks within an RNN. We find that tasks with similar input/output relationships may utilize shared dynamics
in aligned subspaces, even when orthogonality was imposed using our learning rule. For example, when a `Memory`
task is learned first, followed by a `Delay` task with the same input/output structure (e.g. both `Pro` tasks), then the
`Delay` task dynamics can reuse structures of the `Memory` task without changing the input/output relationships in the
`Memory` task subspace. This leads to alignment across both task-subspaces (Figure 6D) and the existence of a trace of a
memory structure (ring attractor) during the performance of the `Delay` task (Figure 7A). **R3** felt that we spent little
effort discussing training order. In the revised paper, we will make it more explicit that analyses of sections 5.3-5.5 were
carried out to better understand training order results (cf lines 164-165). The novel analyses we developed contribute to
a better understanding of learned representations in RNNs and how these are affected by our learning algorithm.

**Transfer learning and orthogonality.** In response to **R4**, we will
update Figure 4 to also include the single task training setting to more
clearly demonstrate transfer learning (see right). **R4** also raised the
question whether orthogonality was imposed in the same way across
experiments. We apply the same learning strategy in all cases, according
to Eq.(3)-(5). We agree that saying "were allowed" in line 195 is

confusing, since the learning algorithm itself is agnostic to any similarities across tasks. **R4** noted that the case where
orthogonality was imposed but the network still used the same subspace, would mean the results of the paper are
nontrivial and interesting, and we would like to note that this was indeed the case. We will revise relevant sections in
the paper to be more explicit about these points.

**Related work and comparisons. R2** pointed out similarities with FORCE, which we refer to in lines 252-256 in the
discussion. We plan to revise the related work section to focus more explicitly on FORCE and also include a reference
to the relevant work in Beer & Barak (2019), as suggested by **R2**. **R3** asked why we focus only on regularization-based
approaches in our comparisons. Our work is motivated by the question of how the *same* neural population may be
involved in computations relating to multiple tasks (lines 23-24). EWC and SI are appropriate for this setting and
represent well-established baselines. Furthermore, SI represents the state-of-the-art on the task-set and architecture we
study here (see Yang, 2019). Replay revisits training data, while we consider the setting where training examples from
previously learned tasks are inaccessible (lines 54-55). Dynamic network architectures (e.g. Li, 2020 and Cossu, 2020)
solve continual learning by growing the network, which is a fundamentally different solution and the reason why we
didn't include such methods as baseline comparisons here. However, we thank **R3** for pointing out the above references
and other recent work on continual learning in the RNN setting. We were not aware of these recent publications, but
agree that they should be discussed as related work in the revised paper.

**Other concerns. R1,3** noted that our set of tasks were limited. We focus on toy examples so that we may analyze the
solutions the network obtains under different training regimes, which we emphasize is a key contribution of our work.
We view this as an important first step, but agree that real-world applications (e.g. cart/pole control in multiple settings,
or brain machine interface control) should be the ultimate goal. **R2** was interested in fixed point structure change
after learning new tasks. Fixed point structures were highly overlapping upon visual inspection in TDR subspaces.
All examined subspace angles between fixed point structures before and after learning subsequent tasks were <0.1
radians, and q values (cf Beer & Barak, 2019) remained in the same range. Such minimal change is consistent with our
learning rule limiting change of dynamics within previously explored subspaces. We will add a discussion of this point
to the revised manuscript. **R2** also raised concerns about biological plausibility and known task boundaries. While the
motivation for our learning rule is based on orthogonal subspace structure in neural populations, the learning rule itself
is not biologically plausible. In the revised manuscript we will clarify that biologically plausible learning, as well as
learning without knowledge of task boundaries are interesting and important directions for future work. Regarding
reproducibility, we will make code publicly available should the paper get accepted.

[Meta-Review · NeurIPS 2020]

The reviewers generally agree that this paper offers a novel viewpoint on avoiding catastrophic forgetting. The theoretical and experimental results are well received. R3 would have preferred to see a deeper discussion on the differences with OWM. However, the authors explained during the rebuttal that their learning rule modifies both sides of the gradient update, differently to OWM. This characteristic, together with the intricacies involved in considering a sequential application, makes the overall contribution significant enough.